# Influence of Dietary Chitosan Feeding Duration on Glucose and Lipid Metabolism in a Diabetic Rat Model

**DOI:** 10.3390/molecules26165033

**Published:** 2021-08-19

**Authors:** Shing-Hwa Liu, Shih-An Feng, Chen-Yuan Chiu, Meng-Tsan Chiang

**Affiliations:** 1Graduate Institute of Toxicology, College of Medicine, National Taiwan University, Taipei 10051, Taiwan; shinghwaliu@ntu.edu.tw; 2Department of Medical Research, China Medical University Hospital, China Medical University, Taichung 40402, Taiwan; 3Department of Pediatrics, College of Medicine, National Taiwan University Hospital, Taipei 10051, Taiwan; 4Department of Food Science, National Taiwan Ocean University, Keelung 20224, Taiwan; afrankgod@gmail.com; 5Center of Consultation, Center for Drug Evaluation, Taipei 115, Taiwan; kidchiou@gmail.com

**Keywords:** chitosan, diabetes, glucose and lipid metabolism

## Abstract

This study was designed to investigate the influence of dietary chitosan feeding-duration on glucose and lipid metabolism in diabetic rats induced by streptozotocin and nicotinamide [a non-insulin-dependent diabetes mellitus (NIDDM) model]. Male Sprague-Dawley rats were used as experimental animals and divided into short-term (6 weeks) and long-term (11 weeks) feeding durations, and each duration contained five groups: (1) control, (2) control + 5% chitosan, (3) diabetes, (4) diabetes + 0.8 mg/kg rosiglitazone (a positive control), and (5) diabetes + 5% chitosan. Whether the chitosan feeding was for 6 or 11 weeks, the chitosan supplementation decreased blood glucose and lipids levels and liver lipid accumulation. However, chitosan supplementation decreased plasma tumor necrosis factor (TNF)-α, insulin levels, alanine aminotransferase (ALT) activity, insulin resistance (HOMA-IR), and adipose tissue lipoprotein lipase activity. Meanwhile, it increased plasma high-density lipoproteins (HDL)-cholesterol level, plasma angiopoietin-like-4 protein expression, and plasma triglyceride levels (at 11-week feeding duration only). Taken together, 11-week (long-term) chitosan feeding may help to ameliorate the glucose and lipid metabolism in a NIDDM diabetic rat model.

## 1. Introduction

Diabetes mellitus (DM) is a global chronic metabolic disease. More than 90% of diabetic patients suffer from type 2 DM (non-insulin-dependent DM; NIDDM) [1]. Type 2 DM is mostly associated with obesity and insulin resistance and may increase the risk of chronic complications such as retinopathy, nephropathy, cardiovascular diseases, and neuropathy [1,2]. Moreover, obesity refers to an excessive amount of adipose tissue, which is an important endocrine organ that secretes adipocytokines such as tumor necrosis factor-alpha (TNF-α), which can enhance insulin resistance and atherosclerosis in diabetic patients [3,4]. 

Dietary fibers are suggested to have beneficial effects on the plasma lipids of patients with diabetes and atherosclerosis. Chitosan is a dietary fiber produced by the deacetylation of chitin. Its chemical structure is similar to cellulose, linked by β (1→4) glycosidic linkage, and is not digested by mammalian digestive enzymes [5]. Chitosan is generally known for its plasma cholesterol-lowering effects due to increased fecal fat excretions [6,7]. The previous studies have also demonstrated that chitosan decreases plasma glucose concentrations through a decrease in liver gluconeogenesis and an increase in skeletal muscle glucose uptake and utilization [8], as well as a reduction in intestinal disaccharidase activity [9] in a streptozotocin (STZ)-induced diabetic rat model. In addition, chitosan has also been shown to reduce plasma TNF-α and adipose weight in the animal models fed a high-fat diet or a cholesterol-enriched diet [10,11]. However, there is little information concerning the hypolipidemic and hypoglycemic effects of chitosan in type 2 diabetes with obesity. Guo et al. (2020) have reported that the blood glucose level of patients with diabetes and obesity/overweight can be improved by supplementation with chitosan for at least 13 weeks [12]. Our previous study revealed that 7-week chitosan feeding did not affect blood glucose level but did decrease HOMA-IR index [13]. These findings indicate that the effects of chitosan on blood glucose and cholesterol concentrations in type 2 diabetes still remain to be clarified. On the other hand, the effectiveness of chitosan on glucose and lipid metabolism for long-term or short-term use is still inconclusive. 

STZ can selectively damage the pancreatic β-cells and develop hypoinsulinemia and hyperglycemia, similar to the type 1 DM condition [14]. Pretreatment of nicotinamide can scavenge free radicals and recruit the consumption of nicotinamide adenine dinucleotide (NAD^+^) in pancreatic β-cells after STZ injection. Hence, the injection of nicotinamide can partially protect the pancreatic β-cells after STZ injection and then develop a NIDDM model with mild hyperglycemia and insulin resistance [14]. 

The present study was to investigate the influence of dietary chitosan [5% high molecular weight (MW) chitosan] feeding duration on glucose and lipid metabolism in a NIDDM rat model induced by streptozotocin and nicotinamide. 

## 2. Results 

### 2.1. Effects of Chitosan on Body Weight, Food Intake, Feed Efficiency, and Organ and Tissue Weights in Diabetic Rats with Different Feeding Duration

First, we tested the influence of chitosan feeding duration on body weight, food intake, feed efficiency, and organ/tissue weights in diabetic rats. After both 6 and 11 weeks of being fed different diets, there were no significant differences in body weight, food intake, and feed efficiency among the groups [control (C), 5% high MW chitosan (CS), diabetes (DM), DM + thiazolidinediones (TZD-rosiglitazone; a positive control), and DM + CS; Table 1]. 

The liver weight/relative liver weight and adipose tissue weight/relative adipose tissue weight were not changed in diabetic rats (DM group) at 6-week feeding duration, but they were significantly increased at 11-week feeding duration, which could be significantly reversed by administration of both chitosan and TZD (Table 2). There was a significant decrease in liver weight/relative liver weight in the CS (versus C), DM + TZD, and DM + CS (versus DM) groups at both 6- and 11-week feeding duration (Table 2). A significant decrease in perirenal fat weight was also observed in the DM + CS group compared to the DM group at both 6- and 11-week feeding durations (Table 2). Compared to the C group, there were significant decreases in liver weight, perirenal fat, and relative weight in the CS group at both 6- and 11-week feeding durations (Table 2).

### 2.2. Effects of Chitosan on Plasma Biochemical Markers and Lipids in Diabetic Rats with Different Feeding Duration

We next examined the influence of chitosan feeding duration on plasma biochemical markers and lipids in diabetic rats. As shown in Table 3, the plasma glucose levels were significantly increased in the DM group at both 6- and 11-week feeding durations, which could be significantly reversed by TZD treatment and chitosan supplementation. For the 6-week feeding duration, there were no significant differences for TNF-α and insulin levels among all groups; except for the DM + TZD group (versus DM), there were no differences among the other groups for the homeostasis model assessment equation-insulin resistance (HOMA-IR) (Table 3). However, there were significant decreases in TNF-α, insulin, and HOMA-IR in the DM + TZD and DM + CS groups (versus DM) at 11-week feeding duration (Table 3). The significant decreases in the TNF-α and insulin levels were also observed in the CS group compared to the C group at 11-week feeding duration only (Table 3). There were significant decreases in plasma ALT and AST activities in DM + TZD and DM + CS groups at 11-week feeding duration compared to the DM group (Table 3). Moreover, plasma ALT activity was significantly decreased in the CS group (versus C group) at the 11-week duration (Table 3). The significant difference between DM + TZD and DM + CS groups for AST activity (at 6-week feeding duration) and TNF-α level (at 11-week feeding duration) was also observed (Table 3).

Moreover, the significant reductions in the levels of the LDL-C, LDL-C + VLDL-C, and HDL-C/ LDL-C + VLDL-C ratios were shown in CS (versus C) and DM + CS (versus DM) groups at both 6- and 11-week feeding duration; but there were significant decreases in the levels of total cholesterol and atherogenic index and the significant increase in the levels of HDL-C and triglyceride in CS (versus C) and DM + CS (versus DM) groups at 11-week feeding duration only (Table 4). The levels of VLDL-C were also decreased in the DM + TZD group (versus DM) at the 11-week feeding duration (Table 4). A significant decrease in the atherogenic index was also shown in the DM + TZD group compared to the DM group at the 11-week feeding duration (Table 4). There was also a decrease in the TC/HDL-C ratio between CS and C groups and a decrease in VLDL-C levels between the DM + TZD and DM groups at a 6-week feeding duration (Table 4). The significant decreases in both total cholesterol and VLDL levels were shown in the DM + CS group compared to the CS group at the 6-week feeding duration (Table 4). There were significant differences between the DM + TZD and DM + CS groups for LDL-C and VLDL-C levels at 6-week feeding duration, and LDL-C and Triglyceride levels and atherogenic index at 11-week feeding duration (Table 4).

### 2.3. Effects of Chitosan on Hepatic and Fecal Lipid Responses in Diabetic Rats with Different Feeding Duration

We next investigated the influence of chitosan feeding duration on hepatic and fecal lipid responses in diabetic rats. As shown in Table 5, there were significant reductions in the levels of liver total cholesterol in the CS (versus C) and DM + CS (versus DM) groups at both 6- and 11-week feeding durations. However, in the DM + TZD group, only the 11-week feeding duration showed a significant decrease in the liver triglyceride levels compared to the DM group. There was a significant decrease in liver total cholesterol levels in the DM + CS group compared to the DM + TZD group on an 11-week feeding duration (Table 5).

Moreover, the fecal weights and levels of fecal total cholesterol and triglyceride were significantly increased in CS (versus C) and DM + CS (versus DM) groups at both 6- and 11-week feeding duration (Table 6). There were significant increases in fecal total cholesterol levels (mg/day) in the 6-week feeding duration group and fecal total cholesterol and triglyceride levels (mg/g feces; mg/day) in the 11-week feeding duration groups in the DM + CS group compared to DM + TZD group (Table 6).

### 2.4. Effects of Chitosan on Metabolic Signaling Proteins in Liver, Plasma and Small Intestine and Enzyme Activities in Adipose Tissue in Diabetic Rats with Different Feeding Duration

We next explored the influence of chitosan feeding duration on metabolic signaling proteins in the liver, plasma, and small intestine enzyme activities in adipose tissue in diabetic rats. As shown in Figure 1 and Figure 2, the levels of protein expression in the livers for phosphorylated AMP-activated protein kinase (AMPK) (pAMPK/AMPK ratio), peroxisome proliferator-activated receptor (PPAR)α, and microsomal triglyceride transfer proteins (MTTP) were significantly increased, and for PPARγ and sterol regulatory element binding protein (SREBP)-1c were significantly decreased in CS (versus C) and DM + CS (versus DM) groups at both 6- and 11-week feeding duration. At 6-week feeding duration, in the DM + TZD group, a significant change in protein expression was observed only for the MTTP protein, which was elevated compared to the DM group; but at the 11-week feeding duration, there were increases in the pAMPK/AMPK ratio, PPARγ, and MTTP protein expression in DM + TZD group compared to the DM group (Figure 1 and Figure 2). There was a decrease in liver PPAR-γ protein expression in the DM + CS group at 11-week feeding duration compared to DM + TZD group (Figure 2). Moreover, the levels of protein expression for plasma angiopoietin-like protein-4 (Angptl4) were not changed in all groups at 6-week feeding duration; but its levels could be significantly increased in CS (versus C) and DM + CS (versus DM) groups at 11-week feeding duration (Figure 3 and Figure 4). The plasma apolipoprotein C3 (ApoCIII) and intestinal mucosal MTTP protein expression levels were significantly decreased, and the intestinal mucosal Angptl 4 protein expression levels were significantly increased in CS (versus C) and DM + CS (versus DM) groups at both 6- and 11-week feeding duration (Figure 3 and Figure 4). There were no changes in the levels of protein expression for Angptl4, ApoCIII, and MTTP in the plasma or intestine in the DM + TZD group (versus DM) at both 6- and 11-week feeding duration (Figure 3 and Figure 4). There was a significant decrease in intestinal MTTP protein expression in the DM + CS group at 11-week feeding duration compared to the DM + TZD group (Figure 4).

The changes of lipoprotein lipase (LPL) activity and lipolysis rate in perirenal adipose tissues were also examined. As shown in Figure 5, the LPL activity was significantly increased in both the DM + TZD and DM + CS groups compared to the DM, DM + CS, and CS groups at 6-week feeding duration; but LPL activity was significantly decreased in CS (versus C) and DM + CS (versus DM) groups, and still increased in the DM + TZD group (versus DM) at an 11-week feeding duration. There was a significant decrease in LPL activity in the DM + CS group at 11-week feeding duration compared to the DM + TZD group (Figure 5). 

Moreover, the lipolysis rate was significantly increased in CS (versus C), DM + TZD (versus DM), and DM + CS (versus DM) at both 6- and 11-week feeding duration (Figure 5). 

## 3. Discussion

Our previous study has shown that both low and high MW chitosan can alleviate hyperglycemia via the inhibition of liver gluconeogenesis and enhancement of skeletal muscle glucose uptake and use in an STZ-induced diabetic rat model [8]. We also found that high MW chitosan had a higher potential than low MW chitosan in reducing hypercholesterolemia and intestinal disaccharidase activity in STZ-induced diabetic rats [9]. We further found that both 5% chitosan oligosaccharide (COS) and 5% high MW chitosan (CS) were capable of improving lipid metabolism in an HF diet-fed rat model through different mechanisms [15]. Our previous study has also shown that CS (high MW) can reduce the plasma fructosamine, leptin, total cholesterol, and HOMA-IR more effectively than low MW chitosan in diabetic rats [13]. In the present study, we tried to confirm the beneficial effect of CS on glucose metabolism and compare the effectiveness of CS on glucose and lipid metabolism for short-term and long-term use in an STZ/nicotinamide-induced diabetic rat model. The present results showed that 11-week (long-term) CS feeding duration was more effective than 6-week (short-term) CS feeding duration in improving glucose and lipid metabolism in diabetic rats.

It has been pointed out that feeding rats with 5% chitosan does not cause changes in food intake or bodyweight in high-fat diet-fed animals [16,17]. The high-fat diet-induced obese rats fed on 5% high molecular weight chitosan for 5–7 weeks showed no body weight change, but the liver weight and the perirenal fat weight could be significantly reduced [18,19]. Moreover, Hsieh et al. (2012) have found that supplementation of 5% high molecular weight chitosan for ten weeks in streptozotocin/nicotinamide-induced diabetic rats does not change the bodyweight but significantly reduced the adipose tissue weight [20]. In the present study, we also found that 5% chitosan supplementation did not affect food intake or bodyweight in diabetic rats. The results of the present study are consistent with these findings. 

TNF-α is involved in insulin resistance in obese animals and can damage insulin function, which down-regulates glucose transporter-4 (GLUT4) expression and action during glucose metabolism and enhances lipolysis [21]. Liu et al. (2010) have shown that high-molecular-weight chitosan can activate Akt (protein kinase B) to promote GLUT4 translocation in the muscle, so that blood glucose can enter cells for metabolic utilization; it can also inhibit the protein expression of phosphoenolpyruvate carboxykinase (PEPCK) and phosphorylated p38 MAPK and increase the phosphorylated AMPK in the liver, indicating that chitosan can inhibit liver gluconeogenesis and enhance skeletal muscle glucose uptake to improve hyperglycemia [8]. In the present study, the 6-week trials showed that the TNF-α concentration did not change significantly in diabetic rats with or without chitosan supplementation. However, in the 11-week trials, the concentrations of TNF-α in diabetic rats were significantly increased, which could be effectively reversed by TZD treatment and chitosan supplementation. Chitosan supplementation could also reduce insulin levels. These results indicate that chitosan supplementation alleviates insulin resistance and regulates insulin secretion, further improving blood glucose levels.

AMPK is an important regulator of metabolism, which can improve energy imbalance by metabolic stress [22]. AMPK can trigger lipid oxidation, increase lipid utilization, and inhibit lipid synthesis and adipocyte differentiation in the liver [22,23,24]. PPARα plays an important role in the gene regulation of lipid metabolism [25]. Studies have shown that PPARα activation in obese and insulin-resistant animal models can effectively stimulate fatty acid oxidation and inhibit the lipid accumulation in liver and muscle tissues, improving insulin sensitivity and insulin resistance [26,27]. PPARγ in the liver plays a role in regulating the homeostasis of triglyceride, leading to hepatic steatosis, but alleviating triglyceride accumulation and insulin resistance in other tissues [28]. In the present study, at both 6- and 11-week feeding duration, the protein expression levels of phosphorylated AMPK and PPARα were significantly increased in the livers of CS and DM + CS groups. In addition, chitosan supplementation could also reduce the protein expression levels of PPARγ and SREBP1c in the livers. These results suggest that chitosan supplementation has the ability to adjust lipid metabolism-related signaling pathways during diabetic conditioning, even in the early stage.

Angiopoietin-like proteins (Angptls) inhibit the activity of LPL, reducing the clearance rate of VLDL [29]. Apo CIII can also regulate LPL activity and control the clearance of VLDL [30]. MTTP exists in cellular microsomes and endoplasmic reticulum and is a lipid transport protein necessary for the synthesis and secretion of VLDL in the liver and chylomicrons in the intestine; MTTP can promote the transport of triglyceride, cholesterol, and phospholipid on both sides of the membrane [31]. In the present study, we found that in the CS and DM + CS groups, the expression of MTTP protein in the liver could be significantly increased, which might increase the release of liver lipids into the blood. Moreover, chitosan supplementation could also reduce the concentration of TNF-α, which might improve insulin resistance, thereby reducing insulin secretion. Yamada et al. (2006) have indicated that insulin can downregulate the expression of Angptl4 in adipocytes [32]. Sukonina et al. (2006) have also reported that the increase of Angptl4 expression can convert the active LPL dimer to an inactive LPL monomer in adipose tissue [33]. Therefore, we speculated that chitosan supplementation reduced the TNF-α level and the insulin secretion after an 11-week feeding, which caused the increase of plasma Angptl4 protein expression and the decrease of LPL activity in the adipose tissue, leading to the increase of plasma triglyceride levels. Gallaher et al. (2002) have also shown that chitosan supplementation can increase plasma triglycerides in normal and diabetic rats [34].

Chiu et al. (2015) have shown that supplementation of 5% and 7% chitosan can increase the expression of MTTP and Apo E and improve the accumulation of lipids in the liver [18]. It has been found that the reduction of Angptl4 expression in the intestine increases pancreatic lipase activity, which can regulate lipid digestion [35]. Studies have shown that in animal models of diabetes or insulin resistance, the MTTP mRNA expression and activity in the intestine are increased, and the content of triglycerides in chylomicrons are increased [36,37,38]. In the present study, we found that chitosan supplementation at both 6- and 11-week feeding durations could improve the accumulation of lipids in the livers of diabetic rats by increasing the protein expression of liver MTTP and inhibit intestinal lipid digestion and absorption capacity by increasing the intestinal Angptl4 protein expression and by decreasing the intestinal MTTP protein expression.

## 4. Materials and Methods

### 4.1. Animals and Experimental Diets

Six-week-old male Sprague-Dawley (SD) rats were purchased from BioLASCO Taiwan Co., Ltd. (Taipei, Taiwan). Rats were housed in cages in the animal room and fed the normal chow diet (Laboratory Rodent Diet 5001, St. Louis, MO, USA) for one week. The temperature of the animal room was maintained at 23 ± 1 °C, the humidity was 40–60%, and the light was controlled for a 12 h light/dark cycle. Food and water were given ad libitum. Rats were randomly divided into five groups (n = 8/group): (1) control (C), (2) control + 5% chitosan (CS), (3) diabetes (DM), (4) diabetes + 0.8 mg/kg TZD (rosiglitazone, p.o.) (DM + TZD), and (5) diabetes + 5% chitosan (DM + CS). For induction of diabetes, nicotinamide (230 mg/kg) and streptozotocin (65 mg/kg in 0.1 M sodium citrate, pH 4.5) were subcutaneously injected into rats, and blood glucose tests were performed one week after induction to determine whether hyperglycemia was induced. The non-diabetic control rats were subcutaneously injected by vehicle (0.5 mL of sodium citrate buffer). The first two groups were non-diabetic groups, and the last three groups were diabetic groups, and they were fed the different experimental diets (the formulation was shown in Table 7) for 6 or 11 weeks. The body weight and food consumption of all groups were measured weekly until euthanasia. Feed efficiency was calculated by [weight gain (g)/food intake (g)] × 100%. This animal study was approved by the Animal House Management Committee of the National Taiwan Ocean University (permission number: 104033). The experimental procedures were in accordance with the Guide for the Care and Use of Laboratory Animals [39]. The determination of the experimental dosage and time course was based on the results of our preliminary tests.

High molecular weight (MW) chitosan was purchased from Koyo Chemical Co. (Tokyo, Japan). The average MW and degree of deacetylation (DD) of chitosan are about 6 × 10^5^ Dalton and 87%, respectively. Thiazolidinediones—rosiglitazone was obtained from Sigma-Aldrich (St. Louis, MO, USA).

### 4.2. Sampling for Blood, Tissues, and Feces

Feces were collected and weighed three days before the euthanasia. The rats fasted for 12 h before euthanasia. Rats were euthanized with carbon dioxide. Blood samples were centrifuged at 3000 rpm for 20 min in a low-speed centrifuge to collect the supernatant, which was the plasma. The liver, adipose tissue (perirenal fat and epididymal fat), and small intestine were collected and weighed. The samples of plasma, liver tissue, and feces were stored at −80 °C until further analysis.

### 4.3. Detection of Triglyceride (TG), Total Cholesterol (TC), and Lipoproteins

The extraction of lipids in the liver and feces was performed as previously described [40]. Briefly, 0.2 g liver or feces in chloroform and methanol mixed solution (2:1, *v*/*v*) were homogenized by a homogenizer and then centrifuged at 1750× *g* for 10 min. The supernatants were collected and stored at −80 °C until further analysis. Both TG and TC levels in the plasma, liver, and feces were determined by enzymatic assay kits (Audit Diagnostics, Cork, Ireland) as previously described [19]. The lipoproteins (HDL, VLDL, and LDL) in the plasma were isolated by density gradient ultracentrifugation as previously described [41]. The atherogenic index in rats was measured with the formula: total cholesterol − HDL cholesterol)/HDL cholesterol [42].

### 4.4. Measurement of Tumor Necrosis Factor-α (TNF-α)

The plasma TNF-α levels were detected by an enzyme-linked immunosorbent assay (ELISA) kit (Assay Designs, Inc., Ann Arbor, MI, USA) for rats, as previously described [20].

### 4.5. Measurement of Plasma Glucose, Insulin, HOMA-IR, Aspartate Aminotransferase (AST), and Alanine Aminotransferase (ALT)

The plasma glucose was detected by a glucose assay kit (Audit Diagnostics). The plasma insulin was measured using an insulin ELISA kit for rats (Mercodia AB, Uppsala, Sweden). HOMA-IR (Homeostasis Model Assessment-Insulin Resistance) = fasting insulin (mU/L) × fasting glucose (mmol/L)/22.5 [43]. The plasma AST and ALT activities were determined using the AST and ALT assay kits (Randox, Antrim, UK). 

### 4.6. Detection of Lipolysis Rate and Lipoprotein Lipase (LPL) Activity

Lipolysis rate was determined by the measurement of glycerol levels as described previously [15]. A glycerol detection kit (Randox, Antrim, UK) was used to assay the levels of glycerol in samples. The nano-moles glycerol/gram adipose tissue/h was used to express the lipolysis rate.

LPL activity was detected by measurement of p-nitrophenol formation as described previously [15]. The absorbance at 400 nm was used for detection.

### 4.7. Western Blotting

The protein extraction and Western blot analysis for protein expression were determined as previously described [19]. Briefly, the proteins of samples were extracted using a radioimmunoprecipitation assay (RIPA) buffer with a cocktail of inhibitors for phosphatase and protease (Thermo Fisher Scientific, Waltham, MA, USA). The concentrations of proteins were detected by a BCA protein assay kit (Thermo Fisher Scientific). For immunoblotting, the equal protein extracts from the liver, plasma, and intestinal mucosa were separated through 8–12% sodium dodecyl sulfate-polyacrylamide electrophoresis (SDS-PAGE) gel. Then, the proteins in the gel were transferred to polyvinylidene difluoride (PVDF) membranes (Bio-Rad, Hercules, CA, USA). We blocked the PVDF membranes for 1 h and then incubated with primary antibodies specific for AMPKα, phosphorylated AMPKα (p-AMPKα) (Cell Signaling Technology, Danvers, MA, USA), Angptl4, PPAR-γ, SREBP1c, PPAR-α, MTTP, ApoCIII, and β-actin (Santa Cruz Biotechnology, Santa Cruz, CA, USA) overnight. Membranes were subsequently probed with horseradish peroxidase-conjugated secondary antibodies. The reactions of antigen and antibody were determined using a Bio-Rad enhanced chemiluminescence kit (Hercules, CA, USA) and exposed to Fujifilm X-ray film (Fujifilm, Tokyo, Japan). The densitometrical quantifications of protein bands were performed by an image software (Image J 1.51; National Institutes of Health, Bethesda, MD, USA).

### 4.8. Statistical Analysis

All results are presented as the mean ± standard deviation (S.D.). The significant differences (*p* < 0.05) among the control and treated groups were analyzed by one-way analysis of variance (ANOVA) and two-tailed Student’s *t*-test with the SPSS statistical software (SPSS, 19.0, Chicago, IL, USA). 

## 5. Conclusions

In this study, we demonstrated that chitosan supplementation for 11 weeks compared to 6 weeks of feeding could significantly reduce the plasma TNF-α, insulin level, ALT activity, HOMA-IR index, and adipose tissue LPL activity. Meanwhile, it increased the plasma HDL-cholesterol level, plasma Angptl4 protein expression, and plasma triglyceride level. The plasma triglyceride levels did not change at the 6-week chitosan feeding duration but significantly increased at the 11-week chitosan feeding duration. This may be due to the decrease of plasma levels of TNF-α and insulin, increased plasma Angptl4 protein, and inhibition of LPL activity in adipose tissue. Moreover, chitosan supplementation could improve the lipid accumulation in the liver and enhance the fecal lipid excretion at both 6- and 11-week feeding durations. Taken together, these findings suggest that 11-week (long-term) dietary chitosan feeding may be more effective in ameliorating glucose and lipid metabolism in a NIDDM diabetic rat model compared to a 6-week feeding duration (short-term use). 

## Figures and Tables

**Figure 1 molecules-26-05033-f001:**
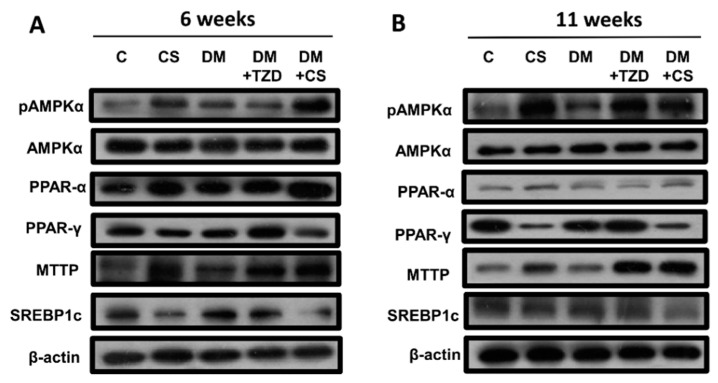
Effects of chitosan on metabolic signaling protein expression in the liver of diabetic rats with different feeding durations. (**A**) 6-week feeding duration; (**B**) 11-week feeding duration. The protein expressions for phosphorylated AMPK, AMPK, PPAR-α, PPAR-γ, MTTP, SREBP1c, and β-actin (internal control) were determined by Western blot analysis. The quantification is shown in Figure 2.

**Figure 2 molecules-26-05033-f002:**
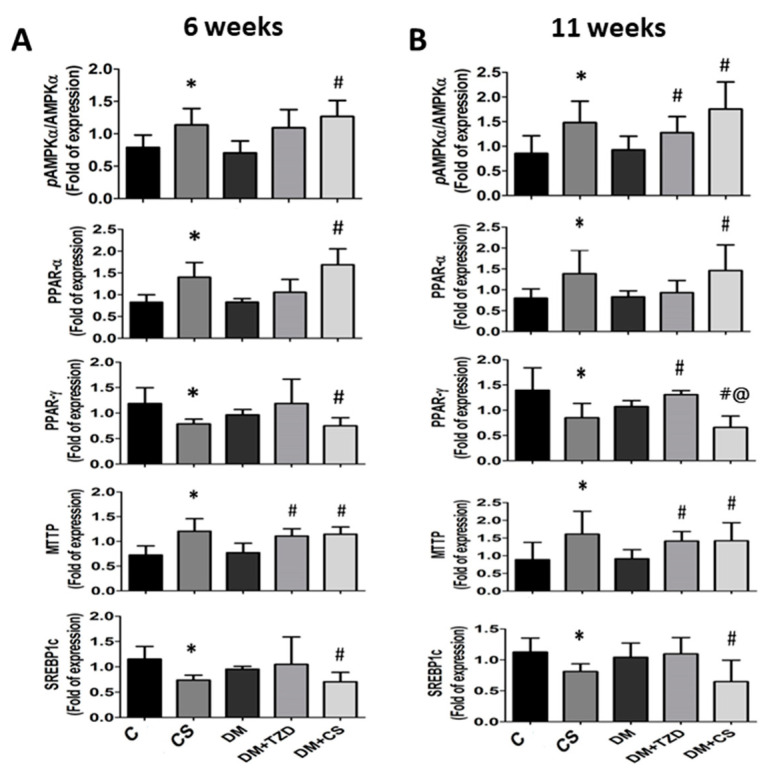
Effects of chitosan on metabolic signaling protein expression in the liver of diabetic rats with different feeding durations. (**A**) 6-week feeding duration; (**B**) 11-week feeding duration. The protein expression for phosphorylated AMPK, AMPK, PPAR-α, PPAR-γ, MTTP, SREBP1c, and β-actin (internal control) was determined by Western blot analysis. The densitometrical quantification is shown. Results are expressed as mean ± S.D. for each group (n = 4–6). The significant difference (*p* < 0.05) were analyzed by one-way ANOVA and two-tailed Student’s *t*-test. *: versus C; #: versus DM; @: versus DM + TZD.

**Figure 3 molecules-26-05033-f003:**
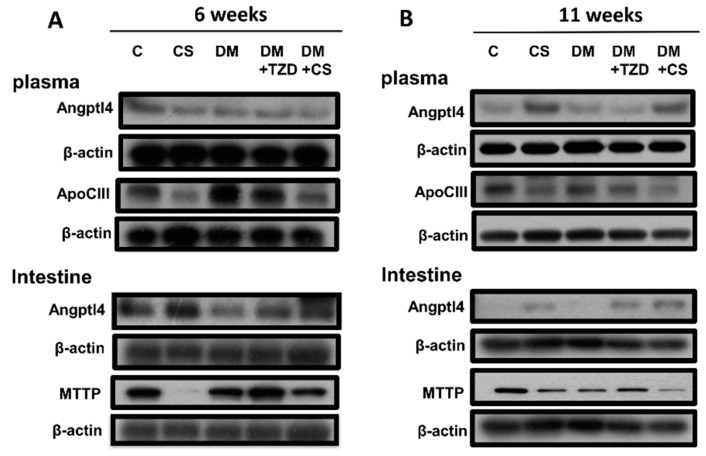
Effects of chitosan on metabolic signaling protein expression in the plasma and intestine of diabetic rats with different feeding durations. (**A**) 6-week feeding duration; (**B**) 11-week feeding duration. The protein expressions for Angptl4, ApoCIII, MTTP, and β-actin (internal control) were determined by Western blot analysis. The quantification is shown in Figure 4.

**Figure 4 molecules-26-05033-f004:**
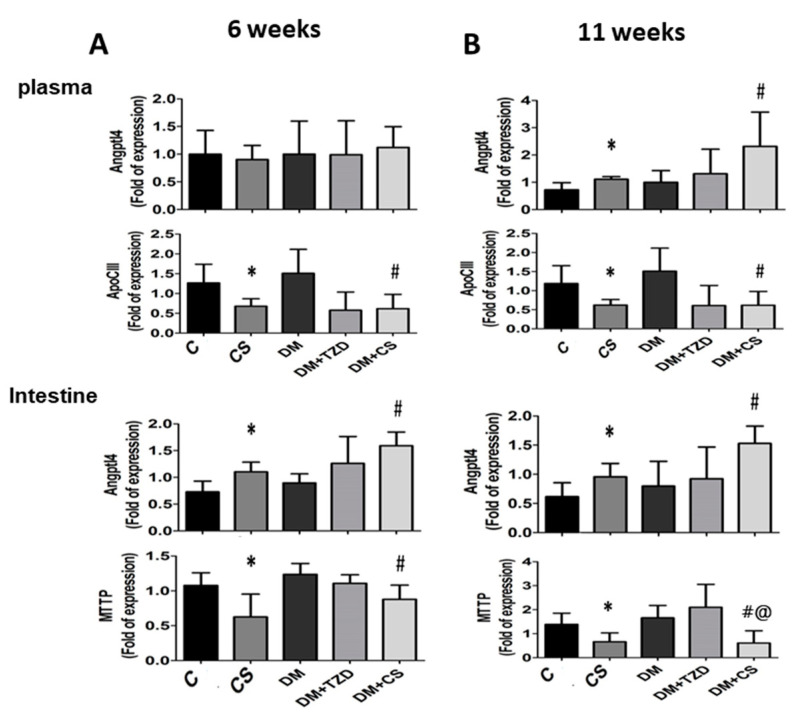
Effects of chitosan on metabolic signaling protein expression in the plasma and intestine of diabetic rats with different feeding durations. (**A**) 6-week feeding duration; (**B**) 11-week feeding duration. The protein expressions for Angptl4, ApoCIII, MTTP, and β-actin (internal control) were determined by Western blot analysis. The densitometrical quantification is shown. Results are expressed as mean ± S.D. for each group (n = 4–6). Significant difference (*p* < 0.05) analyzed by one-way ANOVA and two-tailed Student’s *t*-test. *: versus C; #: versus DM; @: versus DM + TZD.

**Figure 5 molecules-26-05033-f005:**
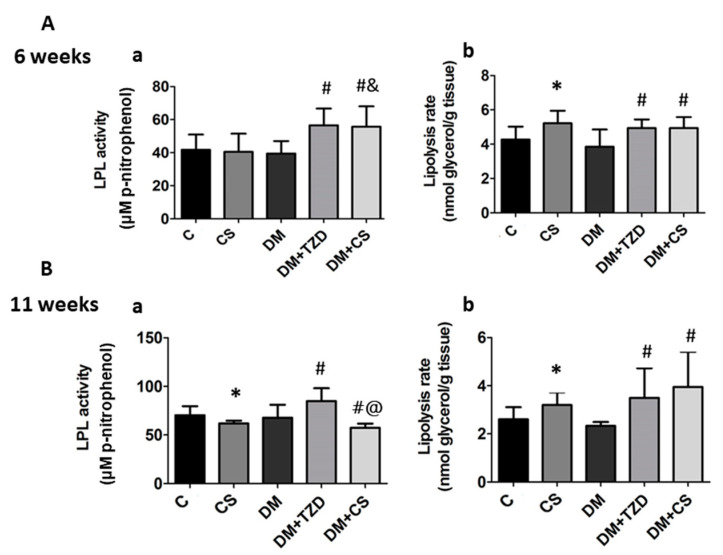
Effects of chitosan on LPL activity and lipolysis rate in the perirenal adipose tissue of diabetic rats with different feeding durations. (**A**) 6-week feeding duration; (**B**) 11-week feeding duration. (**a**) LPL activity; (**b**) lipolysis rate. Results are expressed as mean ± S.D. for each group (n = 8). Significant differences (*p* < 0.05) analyzed by one-way ANOVA and two-tailed Student’s *t*-test. *: versus C; #: versus DM; &: versus CS; @: versus DM + TZD.

**Table 1 molecules-26-05033-t001:** The changes of body weight and food intake in SD rats fed different experimental diets for 6 and 11 weeks.

Diet	C	CS	DM	DM + TZD	DM + CS
**6 weeks**					
Initial body weight (g)	366.7 ± 15.9	366.7 ± 71.8	382.5 ± 14.0	382.1 ± 27.4	385.1 ± 28.4
Final body weight (g)	489.0 ± 28.2	504.1 ± 69.6	523.8 ± 44.1	526.8 ± 41.0	524.5 ± 52.4
Bodyweight gain (g)	122.3 ± 24.0	137.4 ± 49.4	141.3 ± 31.7	144.7 ± 24.1	139.4 ± 27.8
Food intake (g/day)	24.1 ± 2.1	25.5 ± 3.3	27.2 ± 5.6	24.5 ± 4.0	23.3 ± 2.4
Feed efficiency (%)	4.7 ± 0.64	4.7 ± 1.9	4.8 ± 1.2	5.4 ± 1.2	5.6 ± 1.0
**11 weeks**					
Initial body weight (g)	369.8 ± 26.7	368.9 ± 24.6	382.1 ± 16.9	382.7 ± 18.7	383.0 ± 17.8
Final body weight (g)	536.9 ± 37.3	523.8 ± 30.7	555.3 ± 49.9	522.0 ± 40.4	529.0 ± 31.2
Bodyweight gain (g)	170.9 ± 22.9	158.9 ± 29.4	174.2 ± 34.6	143.4 ± 29.9	149.2 ± 26.8
Food intake (g/day)	24.9 ± 1.5	24.3 ± 1.9	26.5 ± 2.5	25.6 ± 2.2	25.6 ± 2.7
Feed efficiency (%)	6.9 ± 0.87	6.7 ± 1.3	6.8 ± 1.7	5.6 ± 0.9	6.0 ± 1.0

Results are expressed as mean ± SD for each group (n = 8). Feed efficiency = [weight gain (g)/food intake(g)] × 100%. C: control diet; CS: control diet +5% high molecular weight chitosan; DM: diabetes + control diet; DM + TZD: diabetes + control diet + 0.8 mg/kg thiazolidinediones (rosiglitazone); DM + CS: diabetes + control diet + 5% high molecular weight chitosan.

**Table 2 molecules-26-05033-t002:** The changes of tissue/organ weights in SD rats fed different experimental diets for 6 and 11 weeks.

Diet	C	CS	DM	DM + TZD	DM + CS
**6 weeks**					
Liver weight (g)	23.4 ± 2.5	19.8 ± 3.3 *	26.8 ± 4.1	22.8 ± 2.7 #	21.6 ± 3.7 #
Relative liver weight (g/100 g BW)	4.9 ± 0.42	4.1 ± 0.44 *	5.2 ± 0.70	4.4 ± 0.42 #	4.2 ± 0.39 #
Perirenal fat (g)	15.9 ± 3.1	12.5 ± 2.7 *	16.7 ± 3.2	14.1 ± 2.3	12.3 ± 3.8 #
Relative perirenal fat weight (g/100 g BW)	3.6 ± 0.74	2.6 ± 0.46 *	3.3 ± 0.60	2.8 ± 0.5	2.4 ± 0.50 #
Epididymal fat (g)	9.5 ± 2.9	8.8 ± 2.7	9.9 ± 2.4	8.8 ± 1.3	9.0 ± 3.0
Total adipose tissue weight (g)	25.4 ± 5.0	21.3 ± 4.6	26.7 ± 4.9	22.9 ± 3.1	21.3 ± 6.7
Relative adipose tissue weight (g/100 g BW)	5.3 ± 1.0	4.4 ± 0.77	5.2 ± 0.80	4.5 ± 0.69	4.1 ± 0.88 #
**11 weeks**					
Liver weight (g)	28.2 ± 3.5	23.1 ± 3.0 *	32.7 ± 3.4 *	25.3 ± 3.1 #	23.1 ± 3.1 #
Relative liver weight (g/100 g BW)	5.3 ± 0.5	4.4 ± 0.49 *	5.9 ± 0.47 *	5.0 ± 0.94 #	4.4 ± 0.43 #
Perirenal fat (g)	14.9 ± 3.4	11.3 ± 3.0 *	18.3 ± 2.9 *	13.2 ± 4.2 #	11.9 ± 3.7 #
Relative perirenal fat weight (g/100 g BW)	2.8 ± 0.6	2.2 ± 0.6	3.3 ± 0.37	2.6 ± 0.80 #	2.3 ± 0.71 #
Epididymal fat (g)	9.0 ± 3.4	8.7 ± 3.6	11.7 ± 1.8	9.0 ± 1.9 #	7.4 ± 1.5 #
Total adipose tissue weight (g)	23.9 ± 5.5	20.1 ± 6.1	30.1 ± 4.3 *	22.2 ± 6.0 #	19.3 ± 5.1 #
Relative adipose tissue weight (g/100 g BW)	4.4 ± 0.87	3.8 ± 1.1	5.4 ± 0.59 *	4.3 ± 1.2 #	3.7 ± 1.0 #

Results are expressed as mean ± SD for each group (n = 8). The significant difference (*p* < 0.05) was analyzed by one-way ANOVA and two-tailed Student’s *t*-test. *: versus C; #: versus DM. C: control diet; CS: control diet +5% high molecular weight chitosan; DM: diabetes + control diet; DM + TZD: diabetes + control diet +0.8 mg/kg thiazolidinediones (rosiglitazone); DM + CS: diabetes + control diet +5% high molecular weight chitosan.

**Table 3 molecules-26-05033-t003:** The changes of plasma AST, ALT, Glucose, insulin concentration, and HOMA-IR index in SD rats fed different experimental diets for 6 and 11 weeks.

Diet	C	CS	DM	DM + TZD	DM + CS
**6 weeks**					
AST (U/L)	121.9 ± 49.8	112.8 ± 40.1	178.5 ± 36.3 *	171.8 ± 59.4	108.0 ± 49.8 #@
ALT (U/L)	30.4 ± 25.9	23.2 ± 13.1	34.6 ± 12.6	28.2 ± 13.7	28.8 ± 10.9
TNF-α (pg/mL)	31.3 ± 10.7	31.7 ± 10.3	42.3 ± 19.9	37.8 ± 20.2	37.4 ± 17.0
Glucose (mg/dL)	190.5 ± 12.2	198.3 ± 11.7	241.6 ± 32.9 *	200.1 ± 24.8 #	203.2 ± 26.5 #
Insulin (μg/L)	0.94 ± 0.25	0.86 ± 0.31	0.88 ± 0.17	0.71 ± 0.30	0.8 ± 0.19
HOMA-IR	9.8 ± 2.4	9.5 ± 3.8	11.9 ± 3.3	8.0 ± 3.6 #	9.3 ± 2.2
**11 weeks**					
AST (U/L)	96.94 ± 60.69	101.7 ± 51.5	205.1 ± 90.45 *	98.2 ± 56.1 #	89.7 ± 52.1 #
ALT (U/L)	30.1 ± 19.8	20.4 ± 4.1 *	48.2 ± 29.4	15.8 ± 4.8 #	20.5 ± 13.1 #
TNF-α (pg/mL)	38.8 ± 9.7	24.0 ± 10.3 *	66.6 ± 24.2 *	18.7 ± 5.0 *#	28.1 ± 9.5 #@
Glucose (mg/dL)	210.7 ± 32.77	218.2 ± 9.44	236.8 ± 14.75 *	206.5 ± 13.55 #	215.6 ± 11.4 #
Insulin (μg/L)	1.1 ± 0.35	0.79 ± 0.13 *	0.92 ± 0.10	0.77 ± 0.12 #	0.76 ± 0.14 #
HOMA-IR	13.6 ± 5.5	9.6 ± 1.9	12.1 ± 1.2	9.3 ± 2.1 #	9.1 ± 2.1 #

Results are expressed as mean ± SD for each group (n = 8). The significant difference (*p* < 0.05) was analyzed by one-way ANOVA and two-tailed Student’s *t*-test. *: versus C; #: versus DM; @: versus DM + TZD. C: control diet; CS: control diet + 5% high molecular weight chitosan; DM: diabetes + control diet; DM + TZD: diabetes + control diet + 0.8 mg/kg thiazolidinediones (rosiglitazone); DM + CS: diabetes + control diet + 5% high molecular weight chitosan.

**Table 4 molecules-26-05033-t004:** The changes of plasma lipids concentration in SD rats fed different experimental diets after 6 and 11 weeks.

Diet	C	CS	DM	DM + TZD	DM + CS
**6 weeks**					
Total cholesterol (mg/dL)	114.8 ± 24.8	98.9 ± 19.5	97.5 ± 17.2	79.5 ± 23.7	81.5 ± 10.5 #&
HDL-C (mg/dL)	40.1 ± 10.9	45.3 ± 12.7	32.5 ± 10.5	33.0 ± 12.2	40.7 ± 12.9
LDL-C (mg/dL)	40.2 ± 15.4	15.3 ± 8.5 *	40.1 ± 29.4	31.9 ± 15.2	13.7 ± 14.4 #@
VLDL-C (mg/dL)	34.5 ± 9.1	38.3 ± 5.8	24.9 ± 6.7	14.6 ± 11.5 #	27.1 ± 10.9 &@
LDL-C + VLDL-C	74.7 ± 19.7	53.7 ± 11.9 *	65.0 ± 26.0	46.5 ± 21.2	40.8 ± 15.9 #
TC/HDL-C ratio	3.0 ± 0.65	2.3 ± 0.39 *	3.5 ± 2.0	2.6 ± 1.1	2.2 ± 0.77
HDL-C/(LDL-C + VLDL-C) ratio	0.57 ± 0.22	0.87 ± 0.25 *	0.61 ± 0.39	0.81 ± 0.33	1.2 ± 0.7 #
Triglyceride (mg/dL)	51.5 ± 13.5	54.9 ± 11.1	39.5 ± 9.7	40.3 ± 12.7	52.4 ± 19.7
Atherogenic index	1.81 ± 0.44	1.26 ± 0.56	2.06 ± 0.39	1.54 ± 0.57	0.94 ± 0.47 #
**11 weeks**					
Total cholesterol (mg/dL)	121.9 ± 21.2	90.1 ± 18.1 *	110.4 ± 31.2	100.4 ± 36.0	80.6 ± 20.1 #
HDL-C (mg/dL)	36.0 ± 10.9	45.8 ± 6.0 *	33.3 ± 9.8	40.5 ± 13.5	44.9 ± 9.8 #
LDL-C (mg/dL)	49.1 ± 22.0	15.0 ± 4.0 *	45.6 ± 29.1	37.3 ± 27.8	12.1 ± 6.6 #@
VLDL-C (mg/dL)	36.7 ± 9.1	29.3 ± 9.7	31.5 ± 6.5	22.5 ± 7.2 #	23.7 ± 11.5
LDL-C + VLDL-C	85.8 ± 25.3	44.3 ± 16.6 *	77.1 ± 30.8	59.8 ± 28.0	35.8 ± 20.8 #
TC/HDL-C ratio	3.8 ± 1.8	2.0 ± 0.30 *	3.6 ± 1.4	2.5 ± 0.63	1.9 ± 0.59 #
HDL-C/(LDL-C + VLDL-C) ratio	0.47 ± 0.25	1.1 ± 0.4 *	0.53 ± 0.36	0.78 ± 0.37	1.8 ± 1.4 #
Triglyceride (mg/dL)	39.4 ± 12.4	58.2 ± 19.7 *	50.5 ± 13.7	31.0 ± 29.3	69.7 ± 14.9 #@
Atherogenic index	1.93 ± 0.49	0.94 ± 0.67 *	2.25 ± 0.58	1.51 ± 0.71 #	0.84 ± 0.43 #@

Results are expressed as mean ± SD for each group (n = 8). The significant difference (*p* < 0.05) was analyzed by one-way ANOVA and two-tailed Student’s *t*-test. *: versus C; #: versus DM; &: versus CS; @: versus DM + TZD. C: control diet; CS: control diet +5% high molecular weight chitosan; DM: diabetes + control diet; DM + TZD: diabetes + control diet + 0.8 mg/kg thiazolidinediones (rosiglitazone); DM + CS: diabetes + control diet +5% high molecular weight chitosan.

**Table 5 molecules-26-05033-t005:** The changes of hepatic lipid profiles in SD rats fed different experimental diets for 6 and 11 weeks.

Diet	C	CS	DM	DM + TZD	DM + CS
**6 weeks**					
Total cholesterol (mg/g liver)	53.7 ± 11.4	34.5 ± 5.2 *	59.5 ± 15.2	45.0 ± 10.7 #	35.9 ± 9.9 #
Triglyceride (mg/g liver)	45.4 ± 11.4	35.6 ± 6.2	59.4 ± 18.2	42.8 ± 12.4	43.7 ± 12.1
**11 weeks**					
Total cholesterol (mg/g liver)	89.5 ± 24.8	57.1 ± 12.7 *	116.4 ± 15.7 *	101.2 ± 12.2	45.8 ± 13.2 #@
Triglyceride (mg/g liver)	63.5 ± 12.3	58.5 ± 9.5	79.4 ± 23.8	52.5 ± 13.4 #	71.7 ± 17.5

Results are expressed as mean ± SD for each group (n = 8). The significant difference (*p* < 0.05) was analyzed by one-way ANOVA and two-tailed Student’s *t*-test. *: versus C; #: versus DM; @: versus DM + TZD. C: control diet; CS: control diet + 5% high molecular weight chitosan; DM: diabetes + control diet; DM + TZD: diabetes + control diet +0.8 mg/kg thiazolidinediones (rosiglitazone); DM + CS: diabetes + control diet + 5% high molecular weight chitosan.

**Table 6 molecules-26-05033-t006:** The changes of fecal weight, total cholesterol, and triglyceride levels in SD rats fed different experimental diets for 6 and 11 weeks.

Diet	C	CS	DM	DM + TZD	DM + CS
**6 weeks**					
Feces wet weight (g/day)	1.9 ± 0.34	2.4 ± 0.37 *	2.0 ± 0.43	2.1 ± 0.18	2.7 ± 0.57 #
Feces dry weight (g/day)	1.7 ± 0.27	2.1 ± 0.34 *	1.7 ± 0.32	1.9 ± 0.15	2.2 ± 0.35 #
Total cholesterol					
(mg/g feces)	17.1 ± 3.4	31.4 ± 8.4 *	17.6 ± 6.5	17.7 ± 3.0	31.4 ± 6.6 #
(mg/day)	28.5 ± 7.3	64.0 ± 17.8 *	31.4 ± 15.2	34.0 ± 7.0	67.6 ± 11.3 #@
Triglyceride					
(mg/g feces)	3.4 ± 0.35	5.1 ± 1.3 *	4.3 ± 0.83	4.9 ± 1.1	6.5 ± 1.4 #
(mg/day)	5.7 ± 1.3	10.4 ± 2.8 *	7.6 ± 2.4	9.2 ± 1.6	14.0 ± 2.4 #&
**11 weeks**					
Feces wet weight (g/day)	1.6 ± 0.24	3.2 ± 0.61 *	1.6 ± 0.34	1.9 ± 0.36	2.9 ± 0.61 #
Feces dry weight (g/day)	1.1 ± 0.26	2.7 ± 0.63 *	1.1 ± 0.31	1.5 ± 0.36	2.3 ± 0.69 #
Total cholesterol					
(mg/g feces)	24.3 ± 6.6	31.7 ± 7.1 *	20.2 ± 5.0	19.8 ± 5.8	29.3 ± 6.2 #@
(mg/day)	26.9 ± 11.3	86.6 ± 27.1 *	23.4 ± 10.5	29.3 ± 12.5	69.0 ± 28.3 #@
Triglyceride					
(mg/g feces)	10.9 ± 1.5	15.0 ± 3.0 *	12.2 ± 2.3	12.8 ± 3.6	15.9 ± 3.3 #@
(mg/day)	12.0 ± 3.5	41.0 ± 11.6 *	13.5 ± 3.5	18.5 ± 5.7	36.5 ± 12.3 #@

Results are expressed as mean ± SD for each group (n = 8). The significant difference (*p* < 0.05) was analyzed by one-way ANOVA and two-tailed Student’s *t*-test. *: versus C; #: versus DM; &: versus CS; @: versus DM + TZD. C: control diet; CS: control diet + 5% high molecular weight chitosan; DM: diabetes + control diet; DM + TZD: diabetes + control diet +0.8 mg/kg thiazolidinediones; DM + CS: diabetes + control diet + 5% high molecular weight chitosan.

**Table 7 molecules-26-05033-t007:** Composition of experimental diets (%).

Ingredient (%)	C	CS	DM	DM + TZD	DM + CS
Casein	20	20	20	20	20
Lard	15	15	15	15	15
Soybean oil	2	2	2	2	2
Vitamin mixture ^1^	1	1	1	1	1
Salt mixture ^2^	4	4	4	4	4
Cholesterol	0.5	0.5	0.5	0.5	0.5
Choline chloride	0.2	0.2	0.2	0.2	0.2
Cholic acid	0.2	0.2	0.2	0.2	0.2
Corn starch	52.1	52.1	52.1	52.1	52.1
Cellulose	5		5	5	
Chitosan ^3^		5			5

C: control diet (15% Lard + 2% Soybean oil). CS: control diet (15% Lard + 2% Soybean oil) + 5% high molecular weight chitosan. DM: diabetes + control diet (15% Lard + 2% Soybean oil). DM + TZD: diabetes + control diet (15% Lard + 2% Soybean oil) + 0.8 mg/kg thiazolidinediones. DM + CS: diabetes + control diet (15% Lard + 2% Soybean oil) + 5% high molecular weight chitosan. ^1^ AIN-93 vitamin mixture. ^2^ AIN-93 mineral mixture. ^3^ The average MW and DD of chitosan about 6 × 10^5^ Dalton and 87%, respectively.

## Data Availability

The data presented in this study are available from the corresponding author upon reasonable request.

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
