# Peer review of "Influence of Dietary Chitosan Feeding Duration on Glucose and Lipid Metabolism in a Diabetic Rat Model"

_molecules, 2021, doi:10.3390/molecules26165033_

Round 1

Reviewer 1 Report

Authors suggest that 11-week (long-term) dietary chitosan feeding may be more effective in ameliorating glucose and lipid metabolism in a NIDDM diabetic rat model than 6-week feeding duration (short-term) use.

Minor Points

1) Table 1: Feed efficiency (%).

Please really explain what you want to measure here. Could you explain what it refers to and how do you measure it? Also, indicate it in the material and method section.

2) Lines 104-105: “There were no significant differences for TNF-α, insulin, and homeostasis model assessment equation-insulin resistance (HOMA-IR)…”.

But the authors do indicate a significant difference for HOMA-IR with DM + TZD (vs DM).

3) Lines 64-67: “The present study was to investigate the influence of dietary chitosan (average MW: 8.6 × 105 dalton) feeding duration on glucose and lipid metabolism in a NIDDM rat model induced by streptozotocin and nicotinamide”.

Is that the molecular weight of chitosan? Or is it 6 x 105 Da as you indicate in the material and method section?

Also, that data is irrelevant in that paragraph. Transfer the information to the material and method section or, in any case, indicate in the paragraph that it is: 5% high molecular weight chitosan.

Major Points

4) Table 2: Small intestine length (cm).

Why did you decide to measure the length of the small intestine? On the other hand, indicate in the text the reason for the increase in length. Could it be an adaptation mechanism against chitosan treatment, by reducing intestinal lipid digestion and absorption? That is, by reducing certain resources (lipids), does it try to compensate it by increasing the reabsorption surface?

Author Response

Reviewer#1

Minor Points:

1) Table 1: Feed efficiency (%). Please really explain what you want to measure here. Could you explain what it refers to and how do you measure it? Also, indicate it in the material and method section.

Response: We appreciate the reviewer's comment. We have added the description for calculation of feed efficiency in the Methods section of this revised manuscript according to the suggestion of reviewer. Feed efficiency was calculated by [weight gain (g) / food intake (g)] x 100%.

2) Lines 104-105: “There were no significant differences for TNF-α, insulin, and homeostasis model assessment equation-insulin resistance (HOMA-IR)…”. But the authors do indicate a significant difference for HOMA-IR with DM + TZD (vs DM).

Response: We appreciate the reviewer's comment. We have corrected the descriptions for this issue in the Results section of this revised manuscript according to the suggestion of reviewer.

3) Lines 64-67: “The present study was to investigate the influence of dietary chitosan (average MW: 8.6 × 105 dalton) feeding duration on glucose and lipid metabolism in a NIDDM rat model induced by streptozotocin and nicotinamide”. Is that the molecular weight of chitosan? Or is it 6 x 105 Da as you indicate in the material and method section? Also, that data is irrelevant in that paragraph. Transfer the information to the material and method section or, in any case, indicate in the paragraph that it is: 5% high molecular weight chitosan.

Response: We appreciate the reviewer's comment. We have corrected the descriptions for this issue in the sections of Introduction and Methods of this revised manuscript according to the suggestion of reviewer.

Major Points

4) Table 2: Small intestine length (cm).

Why did you decide to measure the length of the small intestine? On the other hand, indicate in the text the reason for the increase in length. Could it be an adaptation mechanism against chitosan treatment, by reducing intestinal lipid digestion and absorption? That is, by reducing certain resources (lipids), does it try to compensate it by increasing the reabsorption surface?

Response: We appreciate the reviewer's comment. We agree with the review's opinion for this issue. We observed a phenomenon that chitosan supplementation could increase the small intestine length; however, we did not know the mechanism(s) at the present that need further investigation to understand its mechanism(s) in the future. To avoid confusion, we decided to delete these data for small intestine length and weight.

Reviewer 2 Report

The Authors of this study presented the effect of feeding duration with chitosan on the biochemical parameters in the plasma and liver related to glucose and lipid metabolism in diabetic rats.

I have some remarks regarding the statistical analysis and some fragments in the text.

All my comments and questions are listed below:

  1. I do not understand the sentence in the lines 57-58 “On the other hand, will the effects of chitosan on glucose and lipid metabolism by long-term feeding or short-term feeding is still inconclusive”. Could the Authors rephrase it to be more clear?
  2. In the result section, in the lines 78-79 “they were significantly increased at 11-week feeding duration, which could be significantly reversed by CS and TZD (Table 2)” the Authors used the CS abbreviation for chitosan. The same abbreviation is used for name the non-diabetic group receiving chitosan. Therefore, using the same abbreviation for the substance administered to the rats as well as to name a group is confusing. Maybe it would be better to avoid overusing abbreviations in this case?
  3. The fragment describing the results from table 2 does not mention the changes in liver weight, perirenal fat and their relative weights in the CS groups vs C group. Please add this description.
  4. What is the purpose in weighting the small intestine? Does it change in NIDDM? Or is it connected with the fact that chitosan is dietary fiber?
  5. In the lines 106-107 “but there were the significant decreases in TNF-α, insulin, and HOMA-IR in DM+TZD and DM+CS groups at 11-week feeding duration (Table 3)” and in the lines 109-111 “There were the significant decreases in plasma ALT and AST activities in DM+TZD and DM+CS groups at 11-week feeding duration (Table 3)” there should be added that this was in comparison with the DM group. Moreover, ALT decrease in CS group vs C group at 11 week duration is not mentioned in the main text.
  6. Lines 112-113: In the table 4 there is also a decrease in TC/HDL-C ratio between CS and C groups (week 6) not mentioned in the text. Moreover, the VLDL-C (mg/dL) difference between DM+TZD and DM is not mentioned.
  7. In the paragraph 3.3. the description of statistically significant results for DM+TZG is missing. Also please explain why there are two types of units used in the table 5?
  8. Please add information about the results obtained for DM+TZD group in the paragraph 3.4. Since the Authors used this substance in their research, they should mention it in the results description.
  9. Why only in the figure 5 the DM+CS group is compared statistically with non-diabetic CS group? Are there any other parameters in which such statistically significant differences were observed? If so, please ad this information.
  10. Can the Authors provide statistical comparison between DM+TZD and DM+CS groups? This would show if CS acts stronger or weaker than rosiglitazone.
  11. Regarding the methods. Were non-diabetic rats injected with some kind of buffer or saline (vehicle treated) to obtain the same level or stress as the diabetic groups?
  12. How were the liver and feces prepared in order to determine TG and TC?
  13. How were the samples (protein extracts) prepared for Western Blotting?

Minor remarks:

In the line 241 there is double space between “of triglyceride” words. The same situation is in the line 338 (before the word electrophoresis), 339 (between the brackets and the word gel) and 344 (before the word Membranes)

Author Response

Reviewer#2

Comments and Suggestions for Authors

The Authors of this study presented the effect of feeding duration with chitosan on the biochemical parameters in the plasma and liver related to glucose and lipid metabolism in diabetic rats. I have some remarks regarding the statistical analysis and some fragments in the text. All my comments and questions are listed below:

  1. I do not understand the sentence in the lines 57-58 “On the other hand, will the effects of chitosan on glucose and lipid metabolism by long-term feeding or short-term feeding is still inconclusive”. Could the Authors rephrase it to be more clear?

Response: We appreciate the reviewer's comment. We have rephrased this sentence in the Introduction section of this revised manuscript according to the suggestion of reviewer.

  1. In the result section, in the lines 78-79 “they were significantly increased at 11-week feeding duration, which could be significantly reversed by CS and TZD (Table 2)” the Authors used the CS abbreviation for chitosan. The same abbreviation is used for name the non-diabetic group receiving chitosan. Therefore, using the same abbreviation for the substance administered to the rats as well as to name a group is confusing. Maybe it would be better to avoid overusing abbreviations in this case?

Response: We appreciate the reviewer's comment. We have rephrased this sentence in the Results section of this revised manuscript according to the suggestion of reviewer.

  1. The fragment describing the results from table 2 does not mention the changes in liver weight, perirenal fat and their relative weights in the CS groups vs C group. Please add this description.

Response: We appreciate the reviewer's comment. We have added the description for this issue in the Results section of this revised manuscript according to the suggestion of reviewer.

  1. What is the purpose in weighting the small intestine? Does it change in NIDDM? Or is it connected with the fact that chitosan is dietary fiber?

Response: We appreciate the reviewer's comment. We agree with the review's opinion for this issue. We observed a phenomenon that chitosan supplementation could increase the small intestine length; however, we did not know the mechanism(s) at the present that need further investigation to understand its mechanism(s) in the future. To avoid confusion, we decided to delete these data for small intestine length and weight in this revised manuscript.

  1. In the lines 106-107 “but there were the significant decreases in TNF-α, insulin, and HOMA-IR in DM+TZD and DM+CS groups at 11-week feeding duration (Table 3)” and in the lines 109-111 “There were the significant decreases in plasma ALT and AST activities in DM+TZD and DM+CS groups at 11-week feeding duration (Table 3)” there should be added that this was in comparison with the DM group. Moreover, ALT decrease in CS group vs C group at 11 week duration is not mentioned in the main text.

Response: We appreciate the reviewer's comment. We have added the descriptions for these issues in the Results section of this revised manuscript according to the suggestion of reviewer.

  1. Lines 112-113: In the table 4 there is also a decrease in TC/HDL-C ratio between CS and C groups (week 6) not mentioned in the text. Moreover, the VLDL-C (mg/dL) difference between DM+TZD and DM is not mentioned.

Response: We appreciate the reviewer's comment. We have added the description for this issue in the Results section of this revised manuscript according to the suggestion of reviewer.

  1. In the paragraph 3.3. the description of statistically significant results for DM+TZG is missing. Also please explain why there are two types of units used in the table 5?

Response: We appreciate the reviewer's comment. We have added the description for this issue in the Results section of this revised manuscript according to the suggestion of reviewer. We also revised the Table 5 that only one unit (mg/g liver/) was shown.

  1. Please add information about the results obtained for DM+TZD group in the paragraph 3.4. Since the Authors used this substance in their research, they should mention it in the results description.

Response: We appreciate the reviewer's comment. We have added the description for this issue in the Results section of this revised manuscript according to the suggestion of reviewer.

  1. Why only in the figure 5 the DM+CS group is compared statistically with non-diabetic CS group? Are there any other parameters in which such statistically significant differences were observed? If so, please ad this information.

Response: We appreciate the reviewer's comment. We have added the descriptions for these issues in the Results section of this revised manuscript according to the suggestion of reviewer.

  1. Can the Authors provide statistical comparison between DM+TZD and DM+CS groups? This would show if CS acts stronger or weaker than rosiglitazone.

Response: We appreciate the reviewer's comment. TZD (rosiglitazone) drug is only used as a positive control group to show the rationality of the experimental design and process. Rats were supplemented with chitosan in diets, but rosiglitazone was given orally to rats. Therefore, these two groups may be hard to statistically compare each other.

  1. Regarding the methods. Were non-diabetic rats injected with some kind of buffer or saline (vehicle treated) to obtain the same level or stress as the diabetic groups?

Response: We appreciate the reviewer's comment. The non-diabetic control rats were subcutaneously injected by vehicle (0.5 mL of sodium citrate buffer). We have added this description in the Methods section of this revised manuscript according to the suggestion of reviewer.

  1. How were the liver and feces prepared in order to determine TG and TC?

Response: We appreciate the reviewer's comment. We have added the descriptions for lipid extraction from the liver and feces in order to determine TG and TC in the Methods section of this revised manuscript according to the suggestion of reviewer.

  1. How were the samples (protein extracts) prepared for Western Blotting?

Response: We appreciate the reviewer's comment. We have added the description for protein extraction from the samples for Western blotting in the Methods section of this revised manuscript according to the suggestion of reviewer.

  1. Minor remarks:

In the line 241 there is double space between “of triglyceride” words. The same situation is in the line 338 (before the word electrophoresis), 339 (between the brackets and the word gel) and 344 (before the word Membranes)

Response: We appreciate the reviewer's comment. We have corrected this issue in this revised manuscript according to the suggestion of reviewer.

Reviewer 3 Report

The manuscript entitled «Influence of Dietary Chitosan Feeding Duration on Glucose and Lipid Metabolism in a Diabetic Rat Model» is an interesting work that brings again the attention to a very common product that, in terms of molecular biology, nutrition and food science, has been deeply studied in the recent years. 

The authors should provide and explain the relevance of their work, given the high amount of publications studying similar parameters and conditions, some of their own research group. In example:

  • Liu, S. H., Chen, R. Y., & Chiang, M. T. (2020). Effects of chitosan oligosaccharide on plasma and hepatic lipid metabolism and liver histomorphology in normal Sprague-Dawley rats. Marine Drugs, 18(8), 408.
  • Liu, S. H., Chen, F. W., & Chiang, M. T. (2021). Chitosan Oligosaccharide Alleviates Abnormal Glucose Metabolism without Inhibition of Hepatic Lipid Accumulation in a High-Fat Diet/Streptozotocin-Induced Diabetic Rat Model. Marine Drugs, 19(7), 360.
  • Liu, S. H., Chen, R. Y., & Chiang, M. T. (2021). Effects and Mechanisms of Chitosan and ChitosanOligosaccharide on Hepatic Lipogenesis and Lipid Peroxidation, Adipose Lipolysis, and Intestinal Lipid Absorption in Rats with High-Fat Diet-Induced Obesity. International Journal of Molecular Sciences, 22(3), 1139.

Besides, there are some points that the authors must be very careful to correct and answer, in example:

- L31. Statistical analysis of what data?

- L46. Paraphrase

- L50. Change “little” for the proper word

- L54. Change to “did not”

- L65. MW of chitosan must be indicated when explaining its physical and chemical characteristics. In this line, introduction lacks chitosan properties, and some information about why this product is used.

- Throughout text, please apply deep English style corrections.

- Tables. It would be very useful, even mandatory, to compare results from 6 weeks vs 11 weeks, especially when explaining comparisons like in L105-109.

- L87. Figure? Please check

- Results, L351-354. How was the equality of variance tested? Please include in tables all statistical information. Without that, all data discussion and conclusions remains purely hypothetical. 

- Tables 4-6. Does the calculation of cholesterol content in mass (not detailed in the Material and Methods section) consider both free and esterified cholesterol, with a mean molecular mass for fatty acids esterifying cholesterol?

- The authors did not present the data concerning the effect of chitosan on the atherogenic index. Since it is well known that the atherogenic index of plasma is a strong marker for predicting the coronary artery disease, 

- Were there any dose response studies before using the doses of chitosan? Were there any time response studies before using these two periods?

For all the mentioned above, this reviewer decided to reject this version of the manuscript, unless there are major revision changes.

Author Response

Reviewer#3

Comments and Suggestions for Authors

  1. The manuscript entitled «Influence of Dietary Chitosan Feeding Duration on Glucose and Lipid Metabolism in a Diabetic Rat Model» is an interesting work that brings again the attention to a very common product that, in terms of molecular biology, nutrition and food science, has been deeply studied in the recent years. The authors should provide and explain the relevance of their work, given the high amount of publications studying similar parameters and conditions, some of their own research group. In example: Liu, S. H., Chen, R. Y., & Chiang, M. T. (2020). Effects of chitosan oligosaccharide on plasma and hepatic lipid metabolism and liver histomorphology in normal Sprague-Dawley rats. Marine Drugs, 18(8), 408.; Liu, S. H., Chen, F. W., & Chiang, M. T. (2021). Chitosan Oligosaccharide Alleviates Abnormal Glucose Metabolism without Inhibition of Hepatic Lipid Accumulation in a High-Fat Diet/Streptozotocin-Induced Diabetic Rat Model. Marine Drugs, 19(7), 360.; Liu, S. H., Chen, R. Y., & Chiang, M. T. (2021). Effects and Mechanisms of Chitosan and ChitosanOligosaccharide on Hepatic Lipogenesis and Lipid Peroxidation, Adipose Lipolysis, and Intestinal Lipid Absorption in Rats with High-Fat Diet-Induced Obesity. International Journal of Molecular Sciences, 22(3), 1139.

Response: We appreciate the reviewer's comment. We have added the descriptions for this issue in the Discussion section of this revised manuscript according to the suggestion of reviewer.

  1. Besides, there are some points that the authors must be very careful to correct and answer, in example:

- L31. Statistical analysis of what data?

- L46. Paraphrase

- L50. Change “little” for the proper word

- L54. Change to “did not”

- L65. MW of chitosan must be indicated when explaining its physical and chemical characteristics. In this line, introduction lacks chitosan properties, and some information about why this product is used.

- Throughout text, please apply deep English style corrections.

- Tables. It would be very useful, even mandatory, to compare results from 6 weeks vs 11 weeks, especially when explaining comparisons like in L105-109.

- L87. Figure? Please check

Response: We appreciate the reviewer's comment. We have revised and corrected the descriptions for these issues in this revised manuscript according to the suggestion of reviewer.

  1. - Results, L351-354. How was the equality of variance tested? Please include in tables all statistical information. Without that, all data discussion and conclusions remains purely hypothetical.

Response: We appreciate the reviewer's comment. In the present study, the one-way ANOVA and two-tailed Student’s t-test were used to statistically analyze the significant differences among groups using a commercial SPSS statistical software (SPSS, 19.0, Chicago, IL, USA). As far as we know, in the field of food science, most of the articles do not show the statistical information for equality of variance. We have added some statements for statistical analysis in the table footnotes and figure legends in this revised manuscript.

  1. - Tables 4-6. Does the calculation of cholesterol content in mass (not detailed in the Material and Methods section) consider both free and esterified cholesterol, with a mean molecular mass for fatty acids esterifying cholesterol?

Response: We appreciate the reviewer's comment. In the present study, the TC levels in the samples were determined by a commercial assay kit (Au-dit Diagnostics, Cork, Ireland) that the assay principle of kit to determine total cholesterol was shown as follows:

Cholesterol ester + H2O  cholesterol esterase  cholesterol + fatty acids

Cholesterol + O2   cholesterol oxidase  cholestene-3-one + H2O2 

2H2O2 + 4-aminoantipyrine + phenol  peroxidase   quinoneimine dye +4H2O

  1. - The authors did not present the data concerning the effect of chitosan on the atherogenic index. Since it is well known that the atherogenic index of plasma is a strong marker for predicting the coronary artery disease,

Response: We appreciate the reviewer's comment. We have added the data for atherogenic index in the Table 4 of this revised manuscript according to the suggestion of reviewer.

  1. - Were there any dose response studies before using the doses of chitosan? Were there any time response studies before using these two periods?

Response: We appreciate the reviewer's comment. The determination of the experimental dosage and time course was based on the results of our preliminary tests. We have added this description in the Methods of this revised manuscript.

Round 2

Reviewer 2 Report

The Authors have improved the manuscript according to the majority of my remarks and comments. Nevertheless, there are still some flaws which need to be addressed.

Lines 122-123:

“There was also a decrease in TC/HDL-C ratio between CS and C groups and a decrease in VLDL-C level between DM+TZD and DM groups at 6-week feeding duration (Table 4).” According to the Table 4 VLDL-C decreased in the DM+TZD group also at 11 week od feeding. Please add this information

Lines 167-169:

“At 6-week feeding duration, there was only an increase in the MTTP protein expression in DM+TZD group compared to DM group”

This sentence suggests that MTTP expression was increased only in the DM+TZD group, but not in the DM+CS group at the week 6. Please rewrite this sentence to be more accurate – that at week 6 of the feeding, from all tested genes, in the DM+TZD only the expression for MTTP was elevated and other genes were not affected. E.g. “At 6-week feeding duration, in the DM+TZD group, a significant change in gene expression was observed only for MTTP protein, which was elevated compared to DM group” or something similar.

The response to my previous comment “Why only in the figure 5 the DM+CS group is compared statistically with non-diabetic CS group? Are there any other parameters in which such statistically significant differences were observed? If so, please ad this information” is not satisfactory enough. The Authors did not respond to the question what about other parameters – were DM+CS animals compared to these form the CS group? I have found that triglycerides mg/day at week 6 has statistical “&” marking in the superscript, but it is not mentioned in the results description. Please give a straight answer in the response to my comment – were all parameters obtained for DM+CS compared to these from CS group? If so, which parameters revealed statistically significant differences between DM+CS and CS groups? and add accurate description in the manuscript of such differences .

Regarding my next comment and Authors’ response: “Can the Authors provide statistical comparison between DM+TZD and DM+CS groups? This would show if CS acts stronger or weaker than rosiglitazone.

Response: We appreciate the reviewer's comment. TZD (rosiglitazone) drug is only used as a positive control group to show the rationality of the experimental design and process. Rats were supplemented with chitosan in diets, but rosiglitazone was given orally to rats. Therefore, these two groups may be hard to statistically compare each other.”

I am not convinced – the comparison of these two groups would give information about the potential of chitosan – is it more or less potent than TZD (with regard to the tested parameters)? I do understand that rosiglitazone was a positive control and the route of administration was slightly different (eventually both substances were administered through digestive system), but in my opinion these two groups should also be compared.

Author Response

Reviewer 2:

The Authors have improved the manuscript according to the majority of my remarks and comments. Nevertheless, there are still some flaws which need to be addressed.

  1. Lines 122-123: There was also a decrease in TC/HDL-C ratio between CS and C groups and a decrease in VLDL-C level between DM+TZD and DM groups at 6-week feeding duration (Table 4).” According to the Table 4 VLDL-C decreased in the DM+TZD group also at 11 week od feeding. Please add this information

Response: We appreciate the reviewer's comment. We have added this information in the Results section of this revised manuscript according to the suggestion of reviewer.

  1. Lines 167-169: “At 6-week feeding duration, there was only an increase in the MTTP protein expression in DM+TZD group compared to DM group”

This sentence suggests that MTTP expression was increased only in the DM+TZD group, but not in the DM+CS group at the week 6. Please rewrite this sentence to be more accurate – that at week 6 of the feeding, from all tested genes, in the DM+TZD only the expression for MTTP was elevated and other genes were not affected. E.g. “At 6-week feeding duration, in the DM+TZD group, a significant change in gene expression was observed only for MTTP protein, which was elevated compared to DM group” or something similar.

Response: We appreciate the reviewer's comment. We have revised the description for this issue in the Results section of this revised manuscript according to the suggestion of reviewer.

  1. The response to my previous comment “Why only in the figure 5 the DM+CS group is compared statistically with non-diabetic CS group? Are there any other parameters in which such statistically significant differences were observed? If so, please ad this information” is not satisfactory enough. The Authors did not respond to the question what about other parameters – were DM+CS animals compared to these form the CS group? I have found that triglycerides mg/day at week 6 has statistical “&” marking in the superscript, but it is not mentioned in the results description. Please give a straight answer in the response to my comment – were all parameters obtained for DM+CS compared to these from CS group? If so, which parameters revealed statistically significant differences between DM+CS and CS groups? and add accurate description in the manuscript of such differences .

Response: We appreciate the reviewer's comment. We have re-checked the statistically significant differences between DM+CS and CS groups for all parameters in all Tables and Figures in this revised manuscript according to the suggestion of reviewer. We have revised the descriptions for this issue in the Results section of this revised manuscript.

  1. Regarding my next comment and Authors’ response: “Can the Authors provide statistical comparison between DM+TZD and DM+CS groups? This would show if CS acts stronger or weaker than rosiglitazone.

Response: We appreciate the reviewer's comment. TZD (rosiglitazone) drug is only used as a positive control group to show the rationality of the experimental design and process. Rats were supplemented with chitosan in diets, but rosiglitazone was given orally to rats. Therefore, these two groups may be hard to statistically compare each other.”

I am not convinced – the comparison of these two groups would give information about the potential of chitosan – is it more or less potent than TZD (with regard to the tested parameters)? I do understand that rosiglitazone was a positive control and the route of administration was slightly different (eventually both substances were administered through digestive system), but in my opinion these two groups should also be compared.

Response: We appreciate the reviewer's comment. We have re-checked the statistically significant differences between DM+TZD and DM+CS groups for all parameters in all Tables and Figures in this revised manuscript according to the suggestion of reviewer. We have revised the descriptions for this issue in the Results section of this revised manuscript.